# Long-Term Tailor-Made Exercise Intervention Reduces the Risk of Developing Cardiovascular Diseases and All-Cause Mortality in Patients with Diabetic Kidney Disease

**DOI:** 10.3390/jcm12020691

**Published:** 2023-01-15

**Authors:** Hajime Tamiya, Yuma Tamura, Yasuko Nagashima, Tomoki Tsurumi, Masato Terashima, Kaori Ochiai, Kyosuke Ehara, Tomoki Furuya, Nobuyuki Banba, Yuki Nakatani, Megumi Hoshiai, Asuka Ueno, Takashi Tomoe, Atsuhiko Kawabe, Takushi Sugiyama, Shinya Kawamoto, Takanori Yasu

**Affiliations:** 1Department of Physical Therapy, Niigata University of Health and Welfare, Niigata 950-3198, Japan; 2Department of Rehabilitation, Dokkyo Medical University Nikko Medical Center, Nikko 321-2593, Japan; 3Department of Nursing, Dokkyo Medical University Nikko Medical Center, Nikko 321-2593, Japan; 4Social Participation and Community Health Research Team, Tokyo Metropolitan Institute of Gerontology, Itabashi 173-0015, Japan; 5Department of Physical Therapy, Igaku Academy, Kawagoe 350-0003, Japan; 6Department of Diabetes and Endocrinology, Dokkyo Medical University Nikko Medical Center, Nikko 321-2593, Japan; 7Department of Cardiovascular Medicine and Nephrology, Dokkyo Medical University Nikko Medical Center, 632 Takatoku, Nikko 321-2593, Japan

**Keywords:** cardiovascular events, diabetes mellitus, diabetic kidney disease, exercise, physical therapist

## Abstract

This study aimed to determine the effect of long-term exercise on the risk of developing cardiovascular diseases (CVD) and all-cause mortality in patients with diabetic kidney disease (DKD). A single-center, prospective intervention study using propensity score matching was performed over 24 months. The intervention group (*n* = 67) received six months of individual exercise instruction from a physical therapist, who performed aerobic and muscle-strengthening exercises under unsupervised conditions. New events were defined as the composite endpoint of stroke or CVD requiring hospitalization, initiation of hemodialysis or peritoneal dialysis, or all-cause mortality. The cumulative survival rate without new events at 24 months was significantly higher in the intervention group (0.881, *p* = 0.016) than in the control group (*n* = 67, 0.715). Two-way analysis of variance revealed a significant effect of the group factor on high density lipoprotein-cholesterol (HDL-C) which was higher in the intervention group than in the control group (*p* = 0.004); eGFRcr showed a significant effect of the time factor, which was lower at 24 months than before intervention (*p* = 0.043). No interactions were observed for all items. In conclusion, aerobic exercises combined with upper and lower limb muscle strengthening for six months reduce the risk of developing CVD and all-cause mortality in patients with DKD.

## 1. Introduction

The global diabetic population in 2021 was 536.6 million, estimated to increase to 783.2 million by 2045, with the Western Pacific region, including Japan, reported as having the largest diabetic population worldwide [1]. A multicenter, cross-sectional study in Japan found that 54% of patients with type 2 diabetes mellitus had diabetic kidney disease (DKD) [2]. DKD is a significant cause of end-stage renal disease (ESRD) [3,4,5,6]. Diabetes treatment is associated with high medical costs [7]. In addition, DKD is associated with a high mortality rate and high socioeconomic costs; therefore, there is an urgent need to establish effective strategies to slow the progression of DKD [8].

DKD is strongly affected by cardiovascular diseases (CVD) associated with diabetes mellitus. Notably, diabetes mellitus is an independent and definitive risk factor for CVD [9,10]. Furthermore, between 1990 and 2019, the number of patients with CVD almost doubled from 271 million to 523 million; deaths from CVD increased, accounting for about one-third of global deaths [11]. DKD is more strongly associated with excess risk, as patients with DKD and diabetic retinopathy have a significantly higher risk of death from all-cause mortality and cardiovascular complications [12].

Since the risk of death from CVD increases with the development of DKD [13], preventing CVD may effectively alleviate the severity of DKD. Therefore, increasing physical activity through regular exercise is an essential component of the treatment plan for patients with diabetes. In addition, there is strong evidence of an inverse association between the amount of physical activity and the risk of CVD mortality [14]. Furthermore, moderate- to vigorous-intensity aerobic exercise significantly reduces CVD risk and all-cause mortality in patients with type 1 and type 2 diabetes [15]. While the evidence for exercise in diabetic patients is well established, there is still no consensus on the effectiveness of training for DKD [16,17]. Exercise may be equally effective for DKD patients with underlying diabetes mellitus.

Consequently, we hypothesized that increased physical activity through regular aerobic exercise would be an effective therapeutic tool to reduce the incidence of CVD in patients with DKD. Therefore, this study aimed to determine the impact of increased physical activity through a long-term hybrid exercise program (supervised sessions + home-based sessions) under the guidance of physical DKD therapists on the risk of developing CVD and all-cause mortality.

## 2. Materials and Methods

### 2.1. Study Population

A total of 850 patients registered in our hospital’s outpatient database of diabetes mellitus were included in this study. Of these, 187 cases were excluded, including patients who were difficult to follow up on due to interruption of hospital care, death, and transfer to another hospital and those for whom DKD severity classification was impossible due to lack of data. Of the final 663 participants included in the study, interventions were provided to 89 with consecutive prescriptions for exercise instruction by a physician. Twenty-two patients dropped out during the intervention, and the remaining 67 were classified as the intervention group. A propensity score matching was performed from the historical cohort group based on the basic information of the intervention group and blood sampling data; 67 cases were selected as the control group for this study (Figure 1). Inclusion criteria for the intervention group were type 2 diabetes, DKD (stage I-V), no maintenance dialysis therapy, physician permission for exercise therapy, and no dementia. DKD severity was classified according to eGFR and urinary albumin levels [18] (Appendix A).

### 2.2. Study Design

This is a single-center, prospective interventional study. The observation period was between August 2016 and August 2019; the intervention period was six months. The follow-up period was set as 24 months after the intervention, and the data after 24 months were used for analysis. 

### 2.3. Ethics Approval 

The study was conducted per the Declaration of Helsinki guidelines and the ethics guidelines for clinical research from the Ministry of Health, Labor, and Welfare (Tokyo, Japan). Furthermore, the protocol was reviewed and accepted by the ethics committee of Dokkyo Medical University Nikko Medical Center (Nikko 27001). Written informed consent was obtained from all participants in the intervention group. In addition, informed consent was obtained using the opt-out system in the non-intervention group; those who rejected it were excluded.

### 2.4. Intervention

#### 2.4.1. Educational Programs

The educational session on exercise instruction was designed to help the participants acquire an exercise habit and included diabetes self-management education (DSME) [19]. Because DSME interventions reportedly improve glycemic control [20] and reduce the risk of all-cause mortality [21] with a minimum of 10 h of contact between 6 and 12 months, our educational intervention was conducted following these criteria. In the first month of the intervention, we focused on motivation, including group education, individual education, and counseling; in the second month, the participants were given activity meters and individual exercise instruction and were allowed to exercise unsupervised until the sixth month. During this period of exercise guidance, the patients visited the clinic at least once a month for educational intervention (e.g., benefits and necessity of exercise therapy) and counseling (e.g., listening to factors that inhibit exercise and motivation), confirmation of the content and load of exercise, collection of activity meter data, and measurement of body composition to support the continued exercise.

These educational interventions were conducted by physical therapists certified as cardiac rehabilitation instructors and Certified Diabetes Educators from Japan.

In the control group, the physical therapists did not provide exercise instruction using the DSME. Conversely, diabetic nurses and dietitians provided lifestyle and nutritional guidance to both groups. Therefore, the difference in intervention between the two groups was the physical therapists’ presence or absence of DSME-based exercise instruction.

#### 2.4.2. Exercise Instruction Program

The home-based exercise program consisted of aerobic exercises and eight muscle-strengthening exercises [22,23]. The content of the exercise instruction is as follows;

(1)Walking as a mild-to-moderate aerobic exercise for at least 20 min/day and at least three times/week [24], with a goal of performing this at least 150 min/week [22]. For intensity setting, the target heart rate by the Karvonen method was set according to the stage of each participant (DKD stage I, II: 0.6, III: 0.5, IV: 0.4); if pulse detection was difficult, the Rate of Perceived Exertion by the Borg Scale was set at 12–13 [23].(2)Resistance exercises were performed using TheraBand^®^ and bodyweight resistance for major muscle groups in the upper and lower extremities at least three days per week, with no more than two consecutive days. Participants performed eight resistance exercises (side raise, biceps curl, triceps pull-down, sitting knee raise, knee extension, hamstring curl, and calf raises and squats), with 8–12 repetitions at an intensity that felt somewhat tight (Borg scale: 13) [23,24]. If they could not keep up with these exercises, they were instructed to perform them at an intensity that would not leave them fatigued the next day. In addition, a pamphlet illustrating the contents of the home exercise program was distributed to all participants so they could check the contents at any time.

#### 2.4.3. Encouragement of Physical Activity

To increase adherence to exercise and establish an exercise habit, we measured and recorded the extent of daily physical activity. Consequently, the participants were lent an activity meter (Activity Style Pro HJA-350IT; Omron Healthcare, Kyoto, Japan) and instructed to wear it from when they woke up to when they went to bed for five months, from the second to the sixth month of the exercise instruction intervention. In addition, a self-management notebook was distributed to the participants; they were instructed to record the number of steps, walking time, resistance exercise, weight, and blood pressure at the end of each day.

### 2.5. Physical Activity Measurement

In this study, physical activity was assessed in the intervention group using an activity meter and a short-form of the International Physical Activity Questionnaire (IPAQ) [25].

A 3-axis acceleration sensor was built to measure the amount of physical activity; it can be used with commercially available activity meters [26,27]. The activity meter was worn on the waist for five months (size: 80 × 50 × 20 mm; weight: 60 g including batteries); the participant was instructed to wear it throughout the day except when sleeping and bathing. If the participants wore it for > 10 h per day, the data of that day was adopted; if there were more than two good days that they wore it on weekdays and one day at the weekend, the individual’s data was adopted [28]. The participants were asked to visit the clinic once a month to capture the data from the activity meter; the data were analyzed using 10-s epochs. From the obtained individual data, wear time (min/day), walking time (min/day), and activity time by intensity (min/day) was used for analysis. In addition, the ratio of walking and activity time by intensity was calculated from the recruitment data (min/day/wearing time). These values were not included in the data analysis but were used to ensure that the instructed exercises were performed objectively. We defined light physical activity as activity below 3 METs, moderate-to-vigorous physical activity (MVPA) from 3–6 METs, and vigorous physical activity as activity above 6 METs [24,29]. 

IPAQ estimates were obtained verbally and face-to-face with each participant by trained physical therapists. Physical activity in IPAQ was assessed separately for weekdays and weekends, and these estimates were used to calculate physical activity per week (kcal/week). Physical activity was calculated using the type of activity (indoor, outdoor, etc.), frequency of exercise of 10 min or more (days/week), and duration of exercise (minutes/day) as intensity and type components. Then, METs were calculated and applied to each activity; MET intensity levels used for IPAQ scores were vigorous (8 METs), moderate (4 METs), and walking (3.3 METs; www.ipaq.ki.se (accessed on 10 August 2020)). Using the definition for a MET as the ratio of work metabolic rate to a standard resting metabolic rate of 1.0 (4.184 kJ) ·kg^−1^·h^−1^, 1 MET was considered a resting metabolic rate obtained during quiet sitting [29]. The physical-activity quantity (kcal/week) was calculated from the IPAQ data and weight [30].

### 2.6. Propensity Score Matching

Propensity score matching was used to adjust for differences in background factors between the control and intervention groups [31]. The covariates included in the model were age, gender, body mass index, diabetes history, DKD stage, systolic blood pressure, glycated hemoglobin (HbA1c), hemoglobin (Hb), hi-density lipoprotein cholesterol (HDL-C) and low-density lipoprotein cholesterol (LDL-C). Based on these parameters, the authors aimed to generate pairs of control and intervention groups using logistic regression-based propensity scores. A predefined caliper width of 0.20 was used. The area under the receiver operator acting characteristic curve was drawn to assess the accuracy of the logistic regression model used to calculate the propensity score [32].

### 2.7. Clinical Measurements

The primary endpoint was cumulative survival for the composite endpoint, including new CVD and stroke, hemodialysis induction, and all-cause mortality at 24 months. The definition of CVD was myocardial infarction, ischemic heart disease, heart failure, peripheral arterial disease, and arrhythmia. All of this information was collected from the electronic medical record.

Secondary endpoints included estimated glomerular filtration rates (eGFR), Urinary Albumin/Creatinine ratio (Urinary Alb/Cre ratio), body weight, blood pressure, Triglyceride (TG), High-density lipoprotein cholesterol (HDL-C), LDL-C, DKD Stage, HbA1c, and Hb. The values of these endpoints were obtained from blood and urine collection data during outpatient visits or hospitalization.

eGFR was calculated using the Japanese eGFR equations based on standardized serum creatinine levels as follows: eGFRcr (mL/min/1.73 m^2^) = 194 × serum creatinine − 1.094 × Age − 0.287 (× 0.739 if female) [33]. 

### 2.8. Statistical Analysis

Data are presented as mean ± standard deviation for continuous variables and as numbers and percentages for categorical variables. Baseline comparisons were conducted using the Wilcoxon rank sum test, Student’s t-test, chi-squared test, and Fisher’s exact test. In addition, groups (control, intervention) and by-time (pre, post) repeated-measures analyses of variance were performed to examine interactions. The log-rank test was used to compare the cumulative survival rate against new events within the observation period. COX regression analysis was performed to calculate hazard ratios for CVD occurrence and all-cause mortality risk. JMP ^®^ Pro, Version 14.0 (SAS Institute Inc., Cary, NC, USA) was used for statistical analyses. Statistical significance was set as a two-tailed *p*-value of <5%.

In diabetes, the relative risk of the higher physical activity group is reportedly 0.61 (range: 0.52–0.70) compared with that of the lower physical activity group [34]. In this study, the sample size was calculated according to the report by Kodama [34] and Dupont’s formula [35]. With a significance level of 0.05, a power of 80%, and a median event duration of 24 months [21], the required sample size was 128 cases. Because the dropout rate for intervention studies with follow-up longer than 24 months exceeds 19% [36], we set the dropout rate for this study at 25%.

## 3. Results

The clinical characteristics of the study participants (mean ± SD age, 69 ± 11 years) are shown in Table 1. DKD stage I was represented by 50 cases [37.3%], stage II by 58 cases [43.3%], stage III by 18 cases [13.4%], stage IV by 8 cases [6.0%], and stage V by 0 cases. Ten participants (9.0%) had had a stroke, and 38 (28.4%) had established CVD. When comparing the baseline values of the control and intervention groups, the proportion of users of insulin injections and beta-blockers was significantly higher in the intervention group; however, there were no significant differences in the other items (Table 1, Appendix A).

After 24 months of observation, the cumulative survival rate was significantly higher in the intervention group (0.881) than in the control group (0.715). (Figure 2, *p* = 0.016), with fewer new events. The hazard ratio for combined risk, including all event occurrences, was significantly higher in the control group (HR:2.48, *p* = 0.03). There were 27 cases of new events during the observation period. In the control group, there was 1 case of stroke, 12 of CVD, 1 of HD, and 5 of all-cause mortality. In contrast, in the intervention group, there was one case of stroke, four of CVD, 2 of HD, 1 of all-cause mortality, and fewer cases of CVD and all-cause mortality (Table 2). Hazard ratios for CVD occurrence and all-cause mortality risk were calculated. These results are shown in Table 3. The hazard ratio for the combined risk of CVD and all-cause mortality was significantly higher in the control group (HR:3.70, *p* = 0.01).

Two-way analysis of variance revealed a significant effect of the group factor on HDL-C, which was higher in the intervention group (Group, F (1, 263) = 8.28, *p* = 0.004 vs. matched control); eGFRcr showed a significant effect of the time factor, which was lower at 24 months than before intervention (Time, F1, 260 = 4.12, *p* = 0.043). However, no interaction was observed for all items (Table 4).

### 3.1. Data from Activity Meter

The activity meter data of the intervention group are shown in Table 5. Forty-two of the 67 participants met the inclusion criteria for the activity meter data. During the intervention period, the mean percentage of MVPA to daily wearing time (%) was 9.5 ± 6.1 in the second month, 9.4 ± 6.3 in the third month, 8.8 ± 5.5 in the fourth month, 8.3 ± 5.8 in the fifth month, and 8.7 ± 5.1 in the sixth month.

### 3.2. Data from IPAQ

The amount of physical activity before and during the intervention period is shown in the Appendix A. Compared to the amount of physical activity before the intervention, the amount of physical activity in the third (*p* = 0.0052), fourth (*p* = 0.003), and fifth months (*p* = 0.0013) was significantly higher. 

## 4. Discussion

To our knowledge, this is the first study in which exercise guided by a trained physical therapist decreases the risk of new cardiovascular and all-cause death in patients with DKD. This study’s results contribute to future planning for adequate care and rehabilitation for DKD.

This study showed that increased physical activity through exercise was associated with reduced risk of CVD and all-cause mortality in patients with DKD. The present promising results could have been obtained through consistent instruction provided by the expert physical therapist regarding diabetes management and cardiac rehabilitation instruction [37]. This suggests the potential to expand further the profession of exercise and health professionals with specialized knowledge (regarding pathology, patient education, or exercise). It has been shown that increased physical activity is associated with a significant reduction in CVD risk and all-cause mortality in patients with type 2 diabetes [14,15]; this effect may be similar in patients with DKD. In the intervention group of this study, the ratio of MVPA to the time spent wearing an activity meter per day was approximately 9% (Table 5). Furthermore, since the average time spent wearing an activity meter per day was about 11 h, the participants performed about 415 min of MVPA per week by a simple calculation. The amount of physical activity obtained from the questionnaire shows that the amount of physical activity increased significantly during the intervention period compared to the pre-intervention period (Appendix A). Notably, the participants obtained sufficient physical activity. The fact that the participants in the intervention group achieved sufficient physical activity indicates the usefulness of the DSME-based exercise instruction by a professional physical therapist. In addition, previous studies have shown that patients with diabetes who participated in or continued to participate in more than 10 h of DSME over a 6- to 12-month period had significantly lower mortality and HbA1c than those who spent less time with diabetes care and health education professionals [20,21]. Furthermore, early and tailored progressive rehabilitation interventions have been shown to produce better physical function than usual care [38]. These reports demonstrate the importance of professionals considering the individual’s situation and providing ongoing and tailor-made support. In the present study, HbA1c levels did not change in the intervention group despite adequate physical activity. This may be attributed not only to physical activity, but also to eating habits (quantity, frequency, content, etc.) and medication use. Therefore, these conditions should also be evaluated when implementing rehabilitation.

On the other hand, the 25% dropout rate in this study raises an essential question of how to support these individuals. Notably, there are two main categories of non-participants in the DSME: those who are unable to participate for logistical, medical, or financial reasons (e.g., timing, cost, and existing complications) and those who feel that there is no benefit in participating or who have emotional or cultural concerns (e.g., negative feelings about education that they do not find problematic) [39]. Efforts should be made to identify and address these potential barriers [40]. Therefore, it is an essential future task to analyze the factors that led to dropouts in this study and consider the best approach for those who drop out.

In this study, HDL-C levels were significantly higher in the intervention group at 24 months than in the control. This may be because exercise improved HDL-C levels, which is not easy to obtain with drug treatment, and possibly reduced the risk of CVD and all-cause mortality. Increased physical activity through exercise reportedly prevents atherosclerosis [41] and is a factor in increased blood HDL-C concentration [42]. HDL-C has been reported to increase after ≥120 min of aerobic exercise per week [42]. Since the intervention group in this study could perform approximately 415 min of MVPA per week, we believe that the conditions for increased HDL-C were satisfied. On the other hand, blood glucose and LDL-C levels did not change significantly. This may have been because of the medication. The patients were already well-controlled with statins and hypoglycemic drugs and may have been less affected by exercise.

Carotid artery echo and flow-mediated dilation (FMD) are essential to evaluate blood vessels’ functional and morphological changes due to atherosclerosis. The intima-media thickness obtained by carotid artery echo is closely related to the risk of stroke [43] and CVD [44]; it has been reported that a 1% improvement in FMD reduces the risk of CVD by 13% [45]. Therefore, it is possible that the severity of atherosclerosis also influenced the results of the present study; however, we are not able to address this issue with the present results. To prove this hypothesis, investigating the relationship between vascular assessment using carotid artery echo and FMD and the amount of physical activity would be necessary.

The eGFRcr showed a significant main effect of the time factor and was lower after 24 months in both groups; therefore, we could not show that exercise improved renal function in patients with DKD (Table 4, Appendix A). However, the fact that exercise did not induce decreased renal function or increased urinary Alb/Cre is profound (Table 4, Appendix A). Albuminuria is a useful diagnostic and prognostic indicator of diabetic kidney disease [46]; meta-analysis has shown it to be an independent predictor of CVD and overall mortality [13]. An RCT examining the effects of unsupervised exercise on patients with CKD found that combining aerobic and muscle-strengthening exercises significantly reduced albuminuria better than combining aerobic and balance exercises [47]. The present study’s results suggest that a combined intervention of aerobic and muscle-strengthening exercises does not exacerbate albuminuria in patients with DKD.

An RCT study of exercise and caloric restriction in patients with stage III DKD showed that they reduced inflammation and oxidative stress and delayed renal fibrosis in the intervention group [48]. Low- to moderate-intensity aerobic exercises exert a nephroprotective effect in patients with type 2 diabetes and CKD [49,50]. Conversely, recent advances in treating diabetes and diabetic complications and the aging of patients have increased the number of patients with diabetes who do not follow a typical clinical course. The annual decline in eGFR is 0.7–14.3% in patients with type 2 diabetes, with considerable variability concerning renal function decline [51]. In addition, cystatin C-based eGFR may be more appropriate [52] because eGFR measured with serum creatine is greatly influenced by muscle mass [53]. These findings suggest that the variation in the clinical course of eGFR and changes in muscle mass due to interventions may have influenced eGFR. In the future, it may be necessary to establish an evaluation method that accurately reflects the changes in renal function over time in patients with DKD.

There are some limitations to this study. First, the sample size was limited to a single institution. Therefore, the study can be limited to a small part of the population that can be influenced by different factors. Second, we were not able to assess vascular function. Third, the amount of physical activity in the control group could not be assessed. Finally, given that about 25% of the intervention group dropped out, the study may have been limited to participants with relatively high adherence.

## 5. Conclusions

Exercise in diabetic patients is an effective treatment to reduce the risk of CVD development and death, which may also be effective in DKD. In this study, in patients with DKD, increasing physical activity through a 6-month exercise instruction intervention reduces the risk of developing CVD and all-cause mortality. Our findings indicate that long-term exercise for DKD is an effective means of reducing CVD incidence and all-cause mortality and may be applied to future DKD treatment. 

## Figures and Tables

**Figure 1 jcm-12-00691-f001:**
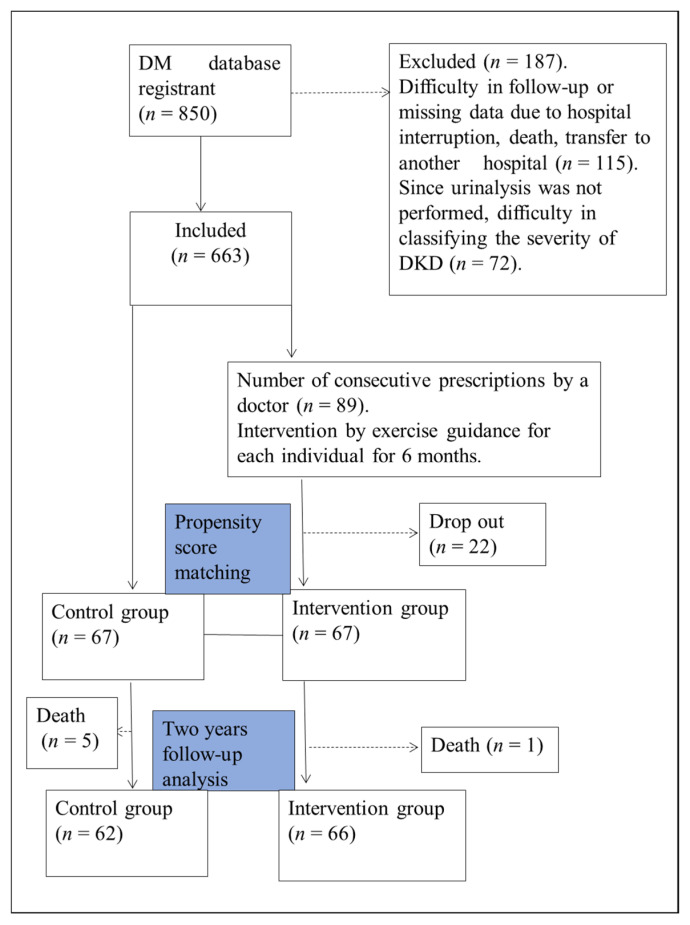
Study flowchart. This is a single-center, prospective intervention study using propensity score matching. The study population comprised 663 patients who could be followed up. Sixty-seven patients who completed the 6-month intervention were designated as the Intervention group. The 67 patients in the cohort group matched by propensity score matching with the intervention group were designated as the control group and followed up for 24 months.

**Figure 2 jcm-12-00691-f002:**
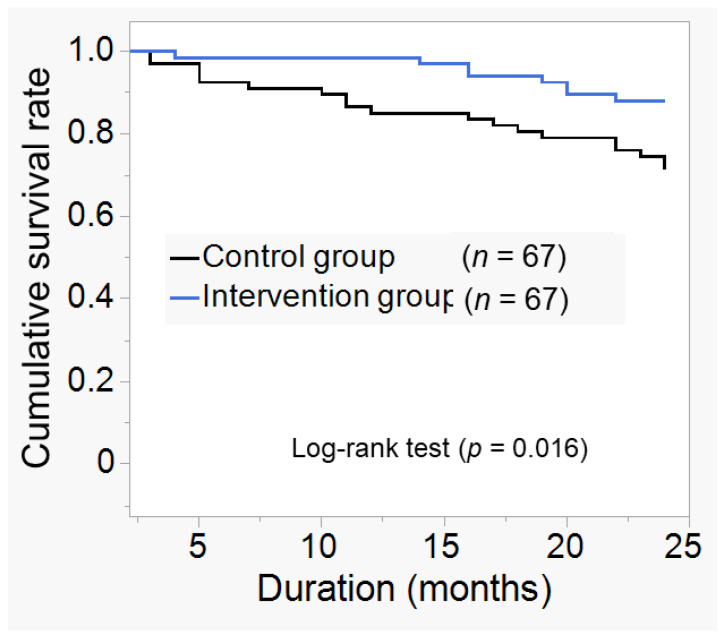
Kaplan–Meier curve for new event incidence. Cumulative survival rates for the composite endpoint included new cardiovascular disease (CVD) and stroke, hemodialysis (HD) induction, and all-cause mortality at 24 months compared to the two groups. The definition of CVD was myocardial infarction, ischemic heart disease, heart failure, peripheral arterial disease, and arrhythmia. After 24 months of observation, the cumulative survival rate for new events was 0.881 in the intervention group and 0.715 in the control group, and was significantly higher in the intervention group (*p* = 0.016).

**Table 1 jcm-12-00691-t001:** Baseline characteristics of participants and non-participants in the intervention before and after propensity score matching.

	Before Propensity Score Matching			After Propensity Score Matching	
	Non-Intervention Group	Intervention Group	*p* Value	Control Group	Intervention Group
	(*n* = 574)	(*n* = 67)		(*n* = 67)	(*n* = 67)
Age (years)	72 ± 12	69 ± 12	0.05 †	69 ± 11	69 ± 12
Male (%)	342 (57.3)	40 (60.0)	0.70 §	40 (60.0)	40 (60.0)
BMI (kg/m^2^)	23.0 ± 3.1	25.8 ± 3.9	<0.01 *†	25.3 ± 3.7	25.8 ± 3.9
Duration of diabetes (years)	10 ± 6	9 ± 5	0.022 *‡	9 ± 5	9 ± 5
DKD Stage (%)			0.53 ¶		
I	280 (47.2)	25 (37.3)		25 (37.3)	25 (37.3)
II	202 (33.8)	29 (43.3)		29 (43.3)	29 (43.3)
III	65 (10.8)	9 (13.4)		9 (13.4)	9 (13.4)
IV	36 (6.0)	4 (6.0)		4 (6.0)	4 (6.0)
V	13 (2.2)	0 (0)		0 (0)	0 (0)
Past History (%)					
Stroke	-	4 (4.5)		5 (7.5)	4 (4.5)
CVD	-	18 (26.9)		19 (28.4)	18 (26.9)
Medication (%)					
insulin injection	86 (14.4)	22 (32.8)	<0.001 *¶	8 (11.9)	22 (32.8)
RAS-Is	318 (53.3)	39 (58.2)	0.61 ¶	37 (55.2)	39 (58.2)
Statin	292 (49.0)	43 (64.2)	0.07 *¶	35 (52.2)	43 (64.2)
SBP (mmHg)	122.1 ± 12.4	129.9 ± 13.5	<0.001 *†	128.0 ± 14.2	129.9 ± 13.5
DBP (mmHg)	61.7 ± 13.5	71.7 ± 9.4	<0.01 *†	69.1 ± 10.6	71.7 ± 9.4
HbA1c (%)	7.0 ± 1.0	6.8 ± 0.8	0.11 †	6.8 ± 0.9	6.8 ± 0.8
Hb (mg/dL)	13.2 ± 1.8	13.2 ± 1.9	0.12 †	13.2 ± 1.9	13.5 ± 1.6
TG (mg/dL)	124.5 ± 34.9	140.6 ± 79.5	0.23 ‡	140.6 ± 96.7	140.6 ± 79.5
HDL-C (mg/dL)	47.1 ± 11.4	51.4 ± 14.2	<0.01 *†	50.2 ± 12.1	51.4 ± 14.2
LDL-C (mg/dL)	105.2 ± 28.0	86.2 ± 23.9	<0.01 *†	85.1 ± 25.8	85.3 ± 21.9
eGFRcr (mL/min/1.73 m^2^)	59.7 ± 21.7	64.6 ± 20.5	0.08 †	62.4 ± 22.3	64.6 ± 20.5
Urinary Alb/Cre ratio	304.7 ± 877.5	234.5 ± 440.7	0.63 ‡	234.8 ± 454.3	234.5 ± 440.7

Data are presented as mean ± standard deviation, or number (%); CVD, Cardiovascular disease; RAS-Is, Renin-Angiotensin system inhibitors; SBP, Systolic blood pressure; DBP, Diastolic blood pressure; G, Triglyceride; HDL-C, High-density lipoprotein cholesterol; LDL-C, Low-density lipoprotein cholesterol; eGFR, estimated glomerular filtration rate; Urinary Alb/Cre ratio, Albumin/creatinine ratio; † Student’s *t*-test; ‡ Wilcoxon rank sum test; § Chi-squared test; ¶ Fisher’s exact test. * *p* < 0.05.

**Table 2 jcm-12-00691-t002:** Breakdown of newly generated events.

	Control Group	Intervention Group
	(*n* = 19)	(*n* = 8)
Event (%)		
Stroke	1 (5.5)	1 (12.5)
Cerebral hemorrhage	0	0
Cerebral infarction	1	1
CVD	12 (70.0)	4 (50.0)
Heart failure	4	2
Ischemic heart disease	5	1
Myocardial infarction	1	1
Arrhythmia	0	0
Peripheral arterial disease	2	0
Hemodialysis	1 (5.5)	2 (25.0)
All-cause mortality	5 (27.8)	1 (12.5)

**Table 3 jcm-12-00691-t003:** Hazard ratio for occurrence of new events for CVD and all-cause mortality combined.

	Hazard Ratio	95% CI (Wald)	*p* Value
CVD	3.03	0.96–9.52	0.058
All cause mortality	2.09	0.38–11.41	0.394
CVD and all-cause mortality	3.70	1.36–10.0	0.01 *

Hazard ratio in the control group. * *p* < 0.05.

**Table 4 jcm-12-00691-t004:** Clinical characteristics between the two groups at the start and end of the observation.

	Baseline		24 Months Later		ANOVA Results					
	Control Group	Intervention Group	Control Group	Intervention Group	Group		Time		Group × Time	
	(*n* = 67)	(*n* = 67)	(*n* = 62)	(*n* = 66)	F	*p* Value	F	*p* Value	F	*p*-Value
BMI (kg/m^2^)	25.3 ± 3.7	25.8 ± 3.9	24.1 ± 7.2	24.5 ± 3.5	0.008	0.93	1.61	0.21	0.81	0.37
SBP (mmHg)	128.0 ± 14.2	129.9 ± 13.4	128.8 ± 15.1	127.9 ± 16.0	0.58	0.46	0.54	0.46	2.99	0.09
DBP (mmHg)	69.1 ± 10.6	71.7 ± 9.3	71.9 ± 14.9	70.2 ± 10.1	0.13	0.72	0.17	0.69	2.16	0.14
HbA1c (%)	6.8 ± 0.9	6.8 ± 0.8	6.9 ± 1.1	6.9 ± 1.0	0.03	0.95	1.18	0.27	0.04	0.84
Hb (mg/dL)	13.2 ± 1.9	13.5 ± 1.6	13.6 ± 1.9	13.3 ± 1.8	0.01	0.92	0.14	0.68	2.30	0.12
TG (mg/dL)	140.6 ± 96.7	140.6 ± 79.5	141.2 ± 91.3	127.2 ± 68.5	0.42	0.51	0.37	0.54	0.44	0.51
HDL-C (mg/dL)	50.2 ± 12.1	51.4 ± 14.1	47.6 ± 16.2	54.9 ± 13.3	5.39	0.02 ***	0.13	0.71	3.33	0.06
LDL-C (mg/dL)	85.1 ± 25.8	85.3 ± 21.9	88.6 ± 25.8	81.8 ± 25.1	1.48	0.22	0.03	0.87	1.72	0.19
eGFRcr (mL/min/1.73 m^2^)	62.4 ± 22.3	64.6 ± 20.1	55.5 ± 23.2	59.0 ± 21.6	1.84	0.17	4.47	0.03 ***	<0.01	0.95
Urinary Alb/Cre ratio	234.8 ± 454.3	234.5 ± 440.7	291.3 ± 612.0	161.9 ± 480.8	0.49	0.48	0.04	0.83	0.77	0.38
DKD Stage (%)										
I	25 (37.3)	25 (37.3)	15 (24.2)	25 (37.9)						
II	29 (43.3)	29 (43.3)	29 (46.8)	24 (36.4)						
III	9 (13.4)	9 (13.4)	8 (12.9)	11 (16.7)						
IV	4 (6.0)	4 (6.0)	9 (14.5)	4 (6.0)						
V	0 (0)	0 (0)	1 (1.6)	2 (3.0)						

Data are presented as mean ± standard deviation, or number (%); Repeated measures two-way ANOVA was performed. * *p* < 0.05.

**Table 5 jcm-12-00691-t005:** Physical activity during the intervention period.

	2nd Month (*n* = 42)	3rd Month (*n* = 42)	4th Month (*n* = 42)	5th Month (*n* = 42)	6th Month (*n* = 42)
Walking time (%) (min/day/wear time)	13.6 ± 6.4	14.1 ± 7.8	13.5 ± 8.0	12.6 ± 7.6	13.3 ± 8.5
LPA (%) (min/day/wear time)	89.2 ± 8.8	88.8 ± 15.7	90.7 ± 6.0	90.6 ± 6.3	91.5 ± 5.3
MVPA (%) (min/day/wear time)	9.5 ± 6.1	9.4 ± 6.3	8.8 ± 5.5	8.3 ± 5.8	8.7 ± 5.1
VPA (%) (min/day/wear time)	1.3 ± 0.3	1.8 ± 0.4	0.5 ± 0.1	1.1 ± 0.2	0.8 ± 0.1
Wear time (min/day)	702.5 ± 134.3	687.2 ± 138.5	674.7 ± 126.3	667.9 ± 135.5	688.4 ± 119.9

LPA: light physical activity (<3 METs), MVPA: moderate-to-vigorous physical activity (3–6 METs), VPA: vigorous physical activity (>6 METs) [24,29]. To measure the amount of physical activity, a 3-axis acceleration sensor that can be used with commercially available activity meters was built in [26,27]. The participant was instructed to wear it throughout the day except when sleeping and bathing; if the participant wore it for more than 10 h per day, the data of that day were adopted, and if there were more than two good days that they wore it on weekdays and one day at the weekend, the individual’s data was adopted [28]. The participants were asked to visit the clinic once a month to capture the data from the activity meter, and the data were analyzed using 10-s epochs. From the obtained individual data, wear time (min/day), walking time (min/day), and activity time by intensity (min/day) was used for analysis. In addition, the ratio of walking time and activity time by intensity was calculated from the recruitment data (min/day/wearing time).

## Data Availability

Not applicable.

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
