# Peer review of "Long-Term Tailor-Made Exercise Intervention Reduces the Risk of Developing Cardiovascular Diseases and All-Cause Mortality in Patients with Diabetic Kidney Disease"

_jcm, 2023, doi:10.3390/jcm12020691_

Round 1

Reviewer 1 Report

The authors show that a 6 month exercise intervention reduces the risk of developing cardiovascular disease and all-cause mortality during 2 yr follow-up in patients with DKD. This protective effect coincided with a moderate increase in HDL-C and less decline in eGFRcr. These are interesting findings that are relevant for supporting patients with DKD. 

Major:

1. The title claims, and the abstract concludes, that the intervention reduces CVD risk and all-cause mortality, but the paper used a composite end point as primary endpoint. From table 2, the difference in CVD and all-cause mortality is not significantly different between the intervention and control group. Is the hazard ratio for CVD and all-cause mortality indeed significantly different, as suggested by the title? Please support the conclusion with these data. 

2. In the intervention group, 22 patients were lost during intervention. Shouldn't those lost cases not have been included in the event-free survival rate analysis? Please show results of this analysis. 

3. It is surprising that insulin injection use was not one of the criteria used in propensity score matching. This suggests that more severe T2DM patients have been included in the intervention group. Might this have affected the results of the intervention? This should at least be discussed.

4. The authors state that the intervention resulted in increased physical activity, yet table 4 only show physical activity of the intervention group during the intervention and not pre-intervention activity levels or activity levels in the control group. Please provide data to support the claim that physical activity is indeed increased during the intervention.

Minor:

Figure 1: correct control group

Table 4: It seems that the table does not present "changes" in physical activity but rather physical activity at different stages of the intervention. 

Line 347: Please consider rephrasing "this cannot be mentioned".

Line 383: please remove the short sentence

Supplementary materials: in the main text a reference to each of these figures is missing. 

Figure S1: what is difference between control group no matching and the non-intervention group before matching?

Author Response

Response to Comments by Reviewer 1

Thank you for carefully reviewing the manuscript entitled “Long-term tailor-made exercise intervention reduces the risk of developing cardiovascular diseases and all-cause mortality in patients with diabetic kidney disease” (manuscript ID: jcm-2080752).

Your comments were very helpful in order to improve our manuscript and provide essential guidance for our research. We hope that our responses and the revised version of our manuscript adequately address your concerns.

Major:

Comment 1:

The title claims, and the abstract concludes, that the intervention reduces CVD risk and all-cause mortality, but the paper used a composite end point as primary endpoint. From table 2, the difference in CVD and all-cause mortality is not significantly different between the intervention and control group. Is the hazard ratio for CVD and all-cause mortality indeed significantly different, as suggested by the title? Please support the conclusion with these data. 

Authors’ Response 1:

According to your suggestion, hazard ratio for the risk of CVD, all-cause mortality, and composite of CVD and all-cause mortality were calculated. As you pointed out, no significant differences between the two groups regarding CVD and all-cause mortality (HR for CVD:3.03, p=0,058, HR for all-cause mortality:2.09, p=0.394, respectively) were found. However, the composite endpoint of CVD events and all-cause mortality confirmed a significant difference (HR for CVD and all-cause mortality: 3.07, p=0.01).This additional result has been added in the text and table 3, as you may see below:

Page 5, lines 228-229

COX regression analysis was performed to calculate hazard ratio for CVD occurrence, all-cause mortality, and composite of CVD and all-cause mortality.

Page 8, lines 265-268

Hazard ratio for CVD occurrence and all-cause mortality risk were calculated. These results are shown in Table 3. The hazard ratio for the combined risk of CVD and all-cause mortality were significantly higher in the control group (HR:3.70, p=0.01).

Page 9

Table 3. Hazard ratio for occurrence of new events for CVD, all-cause mortality, and composite of CVD and all-cause mortality

Hazard ratio

95% CI (Wald)

P value

CVD

3.03

0.96-9.52

0.058

All cause mortality

2.09

0.38-11.41

0.394

CVD and all-cause mortality

3.70

1.36-10.0

0.01*

Comment 2:

In the intervention group, 22 patients were lost during intervention. Shouldn’t those lost cases not have been included in the event-free survival rate analysis? Please show results of this analysis.   

Authors’ Response 2:

In the current study, 22 patients were lost during the intervention and were removed from the event-free survival analysis.

Comment 3:

 It is surprising that insulin injection use was not one of the criteria used in propensity score matching. This suggests that more severe T2DM patients have been included in the intervention group. Might this have affected the results of the intervention? This should at least be discussed. 

Authors’ Response 3:

Thank you for pointing this out. As you noted, there was a significant difference concerning the presence or absence of insulin. This may have influenced the change in HbA1c. Therefore, I have added this to the discussion. Please check below the modification:

Page 11, lines 354-358

In the present study, HbA1c levels did not change . This may be attributed not only to physical activity, but also to eating habits (quantity, frequency, content, etc.) and medication use. Therefore, these conditions should also be evaluated when implementing rehabilitation.

Page 11, lines 375-378

On the other hand, blood glucose and LDL-C levels did not change significantly. This may have been due to the drug. The patients were already well-controlled with statins and hypoglycemic medications and may have been less affected by exercise.

Comment 4:

 The authors state that the intervention resulted in increased physical activity, yet table 4 only show physical activity of the intervention group during the intervention and not pre-intervention activity levels or activity levels in the control group. Please provide data to support the claim that physical activity is indeed increased during the intervention. 

Authors’ Response 4:

Thank you for pointing this out. The amount of physical activity in the intervention group was assessed using a questionnaire and an activity meter. The data obtained from the questionnaire showed a significant increase in physical activity during the intervention period compared to the pre-intervention period. Therefore, this result has been added within the text (as you may see below) and the Supplementary table (Table S1). On the other hand, the amount of physical activity in the control group could not be assessed. Therefore the amount of physical activity in the control group could not be compared with that of the intervention group. This was added as a limitation of the study.

Page 4, lines 166-167

In this study, physical activity was assessed in the intervention group using an activity meter and a short-form of the International Physical Activity Questionnaire (IPAQ)

Page 5, lines 183-193

IPAQ estimates were obtained verbally and face-to-face with each participant by trained physical therapists. Physical activity in IPAQ was assessed separately for weekdays and weekends, and these estimates were used to calculate physical activity per week (kcal/week). Physical activity was calculated using the type of activity (indoor, outdoor, etc.), frequency of exercise of 10 minutes or more (days/week), and duration of exercise (minutes/day) as intensity and type components. Then, METs were calculated and applied to each activity; MET intensity used for IPAQ scores were vigorous (8 METs), moderate (4 METs), and walking (3.3 METs; www.ipaq.ki.se). Using the definition for a MET as the ratio of work metabolic rate to a standard resting metabolic rate of 1.0 (4.184 kJ) *kg-1*h- 1, 1 MET was considered a resting metabolic rate obtained during quiet sitting. The physical-activity quantity (kcal/week) was calculated from the IPAQ data and weight.

Page 10, lines 320-323

The amount of physical activity before and during the intervention period is shown in the Supplementary table (S1). Compared to the amount of physical activity before the intervention, the amount of physical activity in the third (p=0.0052), forth (p=0.003), and fifth months (p=0.0013) was significantly higher. Data are presented as mean ± standard deviation.

Supplementary table

Table S2. Physical activity of IPAQ in the intervention group

Pre intervention(n=67)

2nd month

(n=67)

3rd month

(n=67)

4th month

(n=67)

5th month

(n=67)

6th month

(n=67)

ANOVA

results

Physical activity

(kcal/week)

544.9

±1503.6

1295.1

±1887.2

1708.8

±2162.9*

1750.4

±2043.0*

1632.9

±1880.9*

1344.0

±1644.4

P=0.002

Data are presented as mean ± standard deviation. One-way ANOVA and Tukey HSD were used for statistical analysis. The significance level was set at 5%.

Page 12, lines 415

Third, the amount of physical activity in the control group could not be assessed.

Minor:

Comment 5:

Figure 1: correct control group

Authors’ Response 5:

Thank you for pointing this out. I have corrected it as per your suggestion.

Page 3, lines 96-98

・The word “historical” has been removed.

・The word “Control” has been corrected to “control”.

・Removed the phrase “after the intervention”.

Comment 6:

Table 4: It seems that the table does not present “changes” in physical activity but rather physical activity at different stages of the intervention. 

Authors’ Response 6:

Thank you for pointing this out. The title of Table 5 has been corrected to the version I present below:

Page 10, line 307

Physical activity during the intervention period

Comment 7:

Line 347: Please consider rephrasing "this cannot be mentioned".

Authors’ Response 7:

Thank you for pointing this out. The sentence has been rephrased. Please, see below the modification:

Page 11, lines 384-385

we are not able to address this issue with the present results.

Comment 8

Line 383: please remove the short sentence

Authors’ Response 8:

Thank you for pointing this out. The following sentence has been deleted:

Page 12, line 423

Our findings support.

Comment 9:

Supplementary materials: in the main text a reference to each of these figures is missing. 

Authors’ Response 9:

Thank you for pointing this out. The references have been added:

Page 6, lines 244-25

(Table 1, Figure S1)

Page 10, line 344

(Table S2)

Page 11, line 390

(Table 4, Figure S2)

Page 11, line 391

(Table 4, Figure S3)

Comment 10:

Figure S1: what is difference between control group no matching and the non-intervention group before matching?

Authors’ Response 10:

Thank you for pointing this out. The population from which the matched control group was removed from the non-intervention group became the non-matching control group. I changed it to non-intervention group to improve its readability, as you can check below:

Supplementary Figure S1

The 

Reviewer 2 Report

Dear Authors, 

The research guestion  "does long-term exercise intervention reduces the risk of developing cardiovascular diseases and all-cause mortality in patients with diabetic kidney disease" made me quite courious, and I read this manuscripte with greate interest.  

In this manuscript the gap in knowledge was identified, also the chosen methodology and statistical analysis ensured the validity of the results. Results and interpretation of results are presented appropriately.  

In my opinion, the main limitations are: the small sample size (what you also mentioned in the manuscripte), and not including analysis of the drug that were used.  

Manuscript is clear, relevant for the field and preseted in a well-structured manner. 

I suggest its publishing in present form.

Author Response

Response to Comments by Reviewer 2

Thank you for carefully reviewing the manuscript entitled “Long-term tailor-made exercise intervention reduces the risk of developing cardiovascular diseases and all-cause mortality in patients with diabetic kidney disease” (manuscript ID: jcm-2080752).

Your comments encourage us very much. We hope that the revised version of our manuscript adequately address your concerns.

Reviewer 3 Report

In this study, the authors attempted to elucidate whether a Long-term tailor-made exercise intervention reduces the risk of 3 developing cardiovascular diseases and all-cause mortality in 4 patients with diabetic kidney disease.

In this sense, the positive effect of exercise in the diabetes pathophysiology and symptomatology as well as in the prognosis of the pathology is broadly know. As understood. Authors mainly focus on the DKD, but in the introduction is not well exposed what it is, why is interesting and which kind of the diabetes are they focusing on, because is not the same T1D than T2D. This reviewer did not well understand the hypothesis and the link between CVD and DKD and the novelty of the research.

Methods:

The number of patients included in the study is low. The inclusion criteria is not clear. I understood that only diabetic patients with DKD are included, is it correct?. Are they considering the type of diabetes to inclusion criteria? Which is the criteria for the diagnosis and degree of DKD for the inclusion of the patients? Which stage of the disease are they? Are they taking into account the medication of the patients for inclusion?

It is a single center, so the study is only limited a small part of the population that can be influenced by other factors.

To classify the intensity of the exercise, why don’t[JZ1]  the authors use the heart rate?

Results

Results do not show great significances and are do not show novelty than other studies before.

Discussion

Why is important the increasing in the HDL for the DKD? It is almost the only clinical paramater that changes and we expected that other parameters (LDL, glucose levels, etc) changed, too like other studies show before. Do the authors have any idea about that?

 [JZ1]

Author Response

Response to Comments by Reviewer 3

Thank you for carefully reviewing the manuscript entitled “Long-term tailor-made exercise intervention reduces the risk of developing cardiovascular diseases and all-cause mortality in patients with diabetic kidney disease” (manuscript ID: jcm-2080752).

Your comments were very helpful in order to improve our manuscript and providing essential guidance for our research. We hope that our responses and the revised version of our manuscript adequately address your concerns.

Methods:

Comment 1:

The number of patients included in the study is low.

Authors’ Response 1:

Thank you for pointing this out. Although the population is limited, the sample size was determined by referring to previous studies. As a result, the sample size was calculated to be 64 cases per group, which is the required number for this study.

Comment 2:

The inclusion criteria is not clear. I understood that only diabetic patients with DKD are included, is it correct?. Are they considering the type of diabetes to inclusion criteria? Which is the criteria for the diagnosis and degree of DKD for the inclusion of the patients? Which stage of the disease are they?

Authors’ Response 2:

Thank you for pointing this out. The patients must have type 2 diabetes and DKD (Stage I-V), must not be on maintenance dialysis therapy, must have a physician’s permission for exercise therapy, and must not have dementia. As you indicated, the inclusion criteria could be considered unclear, so we added them to the method section. The severity classification of DKD could also be unclear, so it was added to the Supplementary Table (S1).

Page 2, lines 87-90

Inclusion criteria for the intervention group were type 2 diabetes, DKD (stage I-V), no maintenance dialysis therapy, physician’s permission for exercise therapy, and no dementia. DKD severity was classified according to eGFR and urinary albumin levels (Table S1), as you may see below:

Supplementary Table S1 DKD severity classification

Albuminuria segment

A1

A2

A3

Determination of urinary albumin

Urinary albumin/Creatine ratio (mg / gCr)

Determination of urinary protein
(Urinary protein / creatine ratio)
( g / gCr)

Normal albuminuria
< 30

Microalbuminuria
 30-299

Over albuminuria
≧ 300
or
Highly proteinuria
≧ 0.5

eGFR segment
( ml / min /1.73m2)

≧ 90

Stage I

Stage II

Stage III

60-89

45-59

Stage I

Stage II

Stage III

30-44

15-29

Stage IV

Stage IV

Stage IV

< 15

Stage V

Stage V

Stage V

During dialysis therapy

Stage V

Stage V

Stage V

Comment 3:

Are they taking into account the medication of the patients for inclusion?

Authors’ Response 3:

Thank you for pointing this out. Medications are not considered. As you mentioned, diabetes medication can affect HbA1c results. Therefore, we have added the following comment to the discussion:

Page 11, lines 354-358

In the present study, HbA1c levels did not change despite adequate physical activity in the intervention group. This may be attributed not only to physical activity, but also to eating habits (quantity, frequency, content, etc.) and medication use. Therefore, these conditions should also be evaluated when implementing rehabilitation.

Comment 4:

It is a single center, so the study is only limited a small part of the population that can be influenced by other factors.

Authors’ Response 4:

You are correct. I have added it to the research limitation, as you may see below:

Page 12, lines 412-414

First, the sample size was limited to a single institution. Therefore, the study can be limited to a small part of the population that can be influenced by different factors.

Comment 5:

To classify the intensity of the exercise, why don’t  the authors use the heart rate?

Authors’ Response 5:

Thank you for pointing this out. I apologize for the confusing wording. As you noted, the exercise intensity is determined by setting the target heart rate using the Karvonen method. The coefficients for the Karvonen method were set as DKD stages I and II: 0.6, Stage III: 0.5, and Stage IV: 0.4. For subjects unable to measure their pulse rate, the Borg scale (11-13) was used to set the intensity.

Results

Comment 6:

Results do not show great significances and are do not show novelty than other studies before.

Authors’ Response 6:

Thank you for pointing that out. First, as you noted, no interaction between the blood and urine indices was found. However, a significant difference in the occurrence of the composite endpoints was reported, including the primary outcome of the study, cerebrovascular and cardiovascular disease, and all-cause mortality.

Lastly, this new report demonstrates the effectiveness of exercise instruction interventions provided by physical therapists over an extended period. In addition, long-term exercise did not induce a decrease in eGFRcr or an increase in albuminuria in DKD patients is of great clinical significance. Rehabilitation for DKD is not covered by insurance because no evidence had been previously established. However, we believe that the results of this study contribute to establishing evidence for rehabilitation for DKD and have clinical significance.

The following modifications were performed:

Page 5, lines 228-229

COX regression analysis was performed to calculate hazard ratio for CVD occurrence, all-cause mortality, and composite of CVD and all-cause mortality.

Page 8, lines 265-268

Hazard ratio for CVD occurrence and all-cause mortality risk were calculated. These results are shown in Table 3. The hazard ratio for the combined risk of CVD and all-cause mortality were significantly higher in the control group (HR:3.70, p=0.01).

Page 9

Table 3. Hazard ratio for occurrence of new events for CVD, all-cause mortality, and composite of CVD and all-cause mortality

Hazard ratio

95% CI (Wald)

P value

CVD

3.03

0.96-9.52

0.058

All cause mortality

2.09

0.38-11.41

0.394

CVD and all-cause mortality

3.70

1.36-10.0

0.01*

Discussion

Comment 7:

Why is important the increasing in the HDL for the DKD? It is almost the only clinical paramater that changes and we expected that other parameters (LDL, glucose levels, etc) changed, too like other studies show before. Do the authors have any idea about that?

Authors’ Response 7:

Thank you for pointing this out. We understand that an increase in HDL-C is essential in lowering blood LDL-C and glucose levels and preventing atherosclerosis. In this study, an increase in HDL-C in the intervention group was noticed but no changes in LDL-C concentrations or blood glucose levels were found. This may be because of the medication. The patients were already well-controlled with statins and hypoglycemic drugs, so that they may have been less susceptible to the effects of exercise. This has been added to the discussion, as follows:

Page11, lines 375-378

On the other hand, blood glucose and LDL-C levels did not change significantly. This may have been because of the medication. The patients were already well-controlled with statins and hypoglycemic drugs and may have been less affected by exercise.

Reviewer 4 Report

Very interesting work and with potential for further development in the future. I have just two suggestions for your consideration: 1. Specify further what it means (lines 121-122) "During this period of exercise guidance, the patients visited the clinic at least once a month for educational intervention and counseling"

2. We do not know the professional classes in Japan that are qualified to prescribe and guide physical exercise. In most Western European countries, physiotherapists are not the only professional class with the skills to prescribe this type of exercise. We think that physical education and sports professionals, for example, are the most qualified. Thus, we suggest that in lines 307-308 when it is stated “This suggests the possibility of expanding the profession of physical therapists with specialized knowledge (pathology, patient education, and exercise) in Japan.” Perhaps it would be more appropriate to mention: the possibility of expanding the profession of exercise and health specialists.

Author Response

Response to Comments by Reviewer 4

Thank you for carefully reviewing the manuscript entitled “Long-term tailor-made exercise intervention reduces the risk of developing cardiovascular diseases and all-cause mortality in patients with diabetic kidney disease” (manuscript ID: jcm-2080752).

Your comments were very helpful in order to improve our manuscript and providing essential guidance for our research. We hope that our responses and the revised version of our manuscript adequately address your concerns.

Comment 1:

Specify further what it means (lines 121-122) "During this period of exercise guidance, the patients visited the clinic at least once a month for educational intervention and counseling"

Authors’ Response 1:

Thank you for pointing this out. Educational intervention and counseling content were added in order to ease reader’s comprehension.

Pages 3-4, lines 125-126

During this period of exercise guidance, the patients visited the clinic at least once a month for educational intervention (e.g., benefits and necessity of exercise therapy) and counseling (e.g., listening to factors that inhibit exercise and motivation), confirmation of the content and load of exercise, intake of activity meter data, and measurement of body composition to support the continued exercise.

Comment 2:

We do not know the professional classes in Japan that are qualified to prescribe and guide physical exercise. In most Western European countries, physiotherapists are not the only professional class with the skills to prescribe this type of exercise. We think that physical education and sports professionals, for example, are the most qualified. Thus, we suggest that in lines 307-308 when it is stated “This suggests the possibility of expanding the profession of physical therapists with specialized knowledge (pathology, patient education, and exercise) in Japan.” Perhaps it would be more appropriate to mention: the possibility of expanding the profession of exercise and health specialists.

Authors’ Response 2:

Thank you for your appropriate suggestion. The sentence has been rephrased as follows:

Page 11, line 365-367

This suggests the potential to expand further the profession of exercise and health professionals with specialized knowledge (regarding pathology, patient education, or exercise).

Round 2

Reviewer 1 Report

The authors have addressed my comments adequately. I feel the manuscript has improved significantly.

I have only textual comments:

line 239: mean age does not have an SD; age = 69+/-11 years

Figure 2: please correct contorol in the figure

lines 260-262: please include here the HR for the composite endpoint derived from figure 2

line 266: hazard ratios

line 268: were -> was

line 322: fourth

line 466: please add: during the 6 mo intervention preceding follow-up

Figure S3: please correct urinaly

Author Response

Response to Comments by Reviewer 1

Thank you for carefully reviewing the manuscript entitled “Long-term tailor-made exercise intervention reduces the risk of developing cardiovascular diseases and all-cause mortality in patients with diabetic kidney disease” (manuscript ID: jcm-2080752).

Your comments were very helpful in order to improve our manuscript and provide essential guidance for our research. We hope that our responses and the revised version of our manuscript adequately address your concerns.

Minor comments:

Comment 1:

line 239: mean age does not have an SD; age = 69+/-11 years. 

Authors’ Response 1:

Thank you for pointing this out. I have corrected it as per your suggestion. Please check below the modification:

Page 6, lines 239-240

The clinical characteristics of the study participants (mean ± SD age, 69±11 years) are shown in Table 1.

Comment 2:

Figure 2: please correct contorol in the figure  

Authors’ Response 2:

Thank you for pointing this out; it has been corrected to Control.

Page 8, Figure 2. Kaplan-Meier curve for new event incidence

Comment 3:

lines 260-262: please include here the HR for the composite endpoint derived from figure 2. 

Authors’ Response 3:

Thank you for pointing this out. I have added the hazard ratio, as you suggested. Please check below the modification:

Page 7, lines 262-263

The hazard ratio for combined risk, including all event occurrences, was significantly higher in the control group (HR:2.48, p=0.03).

Comment 4:

line 266: hazard ratios

line 268: were -> was

line 322: fourth

Authors’ Response 4:

Thank you for pointing this out. I have corrected it as per your suggestion. Please check below the modification:

Page 8, line 267

Hazard ratios

Page 8, line 269

was

Page 10, line 323

fourth

Comment 5:

line 466: please add: during the 6 mo intervention preceding follow-up

Authors’ Response 5:

Thank you for pointing this out. I have corrected it as per your suggestion. Please check below the modification:

Page 13, lines 436-437

Table S2. Physical activity of IPAQ in the intervention group during the six months intervention preceding follow-up.

Supplementary table

Table S2. Physical activity of IPAQ in the intervention group during the six months intervention preceding follow-up

Pre intervention (n=67)

2nd month

(n=67)

3rd month

(n=67)

4th month

(n=67)

5th month

(n=67)

6th month

(n=67)

ANOVA

results

Physical activity

(kcal/week)

544.9

±1503.6

1295.1

±1887.2

1708.8

±2162.9*

1750.4

±2043.0*

1632.9

±1880.9*

1344.0

±1644.4

P=0.002

Data are presented as mean ± standard deviation. One-way ANOVA and Tukey HSD were used for statistical analysis. The significance level was set at 5%.

Comment 6:

Figure S3: please correct urinaly

Authors’ Response 6:

Thank you for pointing this out.

Supplementary figure

Figure S3 Changes in urinary Alb/Cre ratio over time
